# OpenReview forum: "Understanding (Un)Reliability of Steering Vectors in Language Models"
_ICLR.cc/2025/Workshop/BuildingTrust — BuildingTrust_

### Official Review · Reviewer_aFfK · 2025-03-01
**SV reliability: Methodological Concerns and Limited Novelty**

**Rating:** 4
**Confidence:** 3

**Review:**

This paper aims to explain some of the unreliability in steering vectors, which are vectors added to certain activations during inference in order to encourage a language model to assume a certain behavior. They analyze prompt type, directional agreement of activations, and separability across projections on the difference-of-means line.

Strengths:
- Addresses an important problem in the field of steering language models, which is the inconsistency and unreliability of steering vectors. The authors validate the existence of this problem with reference to the Tan et al. paper.

- Systematically explores a range of prompt types (instruction, few-shot, prefilled combinations)

Weaknesses:
- Does the use of prefilled prompts undermine the purpose of steering vectors? The study constructs steering vectors using activation differences from prefilled prompts, where the answer token is already appended. However, steering vectors are intended to influence model behavior during inference, not adapt to pre-determined completions. Given this discrepancy, how do the results generalize to real-world inference settings?

- Some results appear somewhat expected: the finding that directional agreement between the steering vector and activation differences predicts steerability seems intuitive, given that the steering vectors themselves are generated using these same activation differences (via the CAA method).

- The "separability" argument, while potentially interesting, lacks sufficient definition and explanation to be fully reproducible -- what exactly is the difference-of-means line? What did you project?

- Lack of novelty: While the paper provides an analysis of steering vector reliability, it doesn't introduce fundamentally new steering methods or techniques. The analysis is more of an investigatory study building upon an existing method (CAA). While such analysis has value, the contribution would be more significant if the authors had demonstrated that their analytical framework generalizes across different steering methods beyond CAA.

---

### Official Review · Reviewer_vQoY · 2025-03-02

**Rating:** 7
**Confidence:** 2

**Review:**

This paper explores the reliability of steering vectors in language models, demonstrating that their effectiveness depends on the target behavior being represented by a consistent linear direction within the model’s activation space. The study reveals that while steering vectors can successfully amplify desired behaviors in some cases, their performance is often unstable and can even degrade outputs, influenced by factors like prompt types and the geometric alignment of activation differences.

The authors’ finding that steering vectors exhibit inconsistent behavior is insightful and helps understand control mechanisms in language models. The study shows that success relies on specific conditions, such as high directional agreement and separability in the activation space. This provides a clearer view of the technique and a starting point for future improvements.

As mentioned in the paper, experiments on broader datasets, more comprehensive steering vector methods, and more recent language models would clarify if the findings extend further. That being said, this paper offers a detailed analysis of steering vector reliability and adds to our knowledge of activation-based interventions in language models. Its findings are relevant and provide a foundation for future studies. Based on its merits, I suggest accepting this paper for publication.

---

### Decision · Program_Chairs · 2025-03-02

Accept